# The Effect of Escitalopram on Central Serotonergic and Dopaminergic Systems in Patients with Cervical Dystonia, and Its Relationship with Clinical Treatment Effects: A Double-Blind Placebo-Controlled Trial

**DOI:** 10.3390/biom10060880

**Published:** 2020-06-08

**Authors:** Evelien Zoons, Marina A.J. Tijssen, Yasmine E.M. Dreissen, Marenka Smit, Jan Booij

**Affiliations:** 1Department of Neurology, Zaans Medisch Centrum, 1502 DV Zaandam, The Netherlands; zoons.e@zaansmc.nl; 2Department of Neurology, University Medical Centre, 9713 GZ Groningen, The Netherlands; m.a.j.de.koning-tijssen@umcg.nl (M.A.J.T.); m.smit03@umcg.nl (M.S.); 3Department of Neurosurgery, Amsterdam University Medical Centre, Location Academic Medical Centre, 1100 DD Amsterdam, The Netherlands; y.e.dreissen@amsterdamumc.nl; 4Department of Radiology and Nuclear Medicine, Amsterdam UMC, Location Academic Medical Centre, 1100 DD Amsterdam, The Netherlands

**Keywords:** cervical dystonia, single-photon emission computed tomography (SPECT), dopamine serotonin, escitalopram

## Abstract

*Purpose:* The pathophysiology of cervical dystonia (CD) is thought to be related to changes in dopamine and serotonin levels in the brain. We performed a double-blind trial with escitalopram (selective serotonin reuptake inhibitor; SSRI) in patients with CD. Here, we report on changes in dopamine D_2/3_ receptor (D2/3R), dopamine transporter (DAT) and serotonin transporter (SERT) binding potential (BP_ND_) after a six-week treatment course with escitalopram or placebo. *Methods:* CD patients had [123I]FP-CIT SPECT (I-123 fluoropropyl carbomethoxy-3 beta-(4-iodophenyltropane) single-photon emission computed tomography) scans, to quantify extrastriatal SERT and striatal DAT, and [123I]IBZM SPECT (I-123 iodobenzamide SPECT) scans to quantify striatal D2/3R BPND before and after six weeks of treatment with either escitalopram or placebo. Treatment effect was evaluated with the Clinical Global Impression scale for dystonia, jerks and psychiatric symptoms, both by physicians and patients. *Results:* In both patients treated with escitalopram and placebo there were no significant differences after treatment in SERT, DAT or D2/3R BP_ND_. Comparing scans after treatment with escitalopram (n = 8) to placebo (n = 8) showed a trend (*p* = 0.13) towards lower extrastriatal SERT BPND in the SSRI group (median SERT occupancy of 64.6%). After treatment with escitalopram, patients who reported a positive effect on dystonia or psychiatric symptoms had significantly higher SERT occupancy compared to patients who did not experience an effect. *Conclusion:* Higher extrastriatal SERT occupancy after treatment with escitalopram is associated with a trend towards a positive subjective effect on dystonia and psychiatric symptoms in CD patients.

## 1. Introduction

Cervical dystonia (CD) is the most common form of adult-onset, isolated, focal dystonia [1]. Besides dystonia of the neck muscles, patients often suffer from a variety of other symptoms, such as jerks or head tremor (around 50% of patients) [2] and psychiatric symptoms, mainly depression and anxiety (lifetime prevalence between 40–90%) [3,4].

The pathophysiology of CD and other forms of dystonia is not fully understood, but alterations in neurotransmitters such as dopamine and serotonin are hypothesized to play an important role [5,6]. In line with this hypothesis, in two recent single-photon emission computed tomography (SPECT) studies in CD, we have shown that abnormalities in central dopaminergic and serotonergic systems are mainly related to depressive symptoms [7,8].

Treatment with selective serotonin re-uptake inhibitors (SSRIs) is an effective pharmacological intervention for major depression [9]. After short-term treatment with SSRIs (4–6 weeks), serotonin transporter (SERT) binding in midbrain, thalamus and striatum is significantly reduced (40–70%) in depressed patients as well as healthy controls, representing occupancy of the SERT by the SSRI [10,11,12]. Treatment of major depression with a therapeutic dose of different SSRIs led to midbrain SERT occupancy of around 80%, if measured shortly after ingestion of the last SSRI dosage [13,14,15]. Increasing the SSRI dose leads to higher plasma levels but SERT occupancy does not further increase (ceiling effect). This is in line with the fact that increasing the dose of an SSRI does not increase treatment efficacy [16], but may lead to more side effects, likely (partly) caused by an effect of SSRIs on other neurotransmitter systems, such as the dopamine system. It was found that the striatal DAT binding is significantly increased by approximately 20% during a successful six-week treatment course with an SSRI, although this effect is smaller in older patients [11].

In dystonia, the role of the serotonin system is far less clear than in major depression. However, serotonergic abnormalities are likely to play a role in the pathophysiology of CD (for a recent review see Smit and co-workers) [6]. Recently, we conducted a crossover trial with the SSRI escitalopram, to evaluate the efficacy in CD with jerks/tremor. In this particular study, we did not find an add-on SSRI treatment effect to botulinum neurotoxin (BoNT) comparing a six-week treatment course with escitalopram or placebo [17]. To examine SERT occupancy, and the relationship between SERT occupancy and clinical effect, as well as to evaluate the effect of SSRI treatment in CD on the dopaminergic system, we studied the effect on both the dopamine and serotonin system in each subject, before and after treatment with serial SPECT scans. In the current article, we report the effects on DAT, D_2/3_ receptors (D2/3R) and SERT BP_ND_, as measured with SPECT, in relation to the clinical effect. We hypothesized that in CD patients treated with placebo tracer binding to DAT, D2/3R and SERT would remain unchanged. In CD patients treated with escitalopram, we hypothesized that SERT binding would be significantly reduced (representing occupancy, expected around 80%) and D2/3R and DAT binding would be increased. Furthermore, we investigated whether SERT occupancy and change in D2/3R and DAT binding were related to clinical outcomes.

## 2. Material and Methods

### 2.1. Subjects

Inclusion and exclusion criteria have been published previously [7]. In short, patients were eligible to participate if they were between 35 and 80 years old and had idiopathic CD that had been clinically stable for at least one year on the Tsui scale for dystonia severity with three-times-monthly BoNT injections. Exclusion criteria were other neurological conditions at inclusion or in the past, treatment with deep brain stimulation for dystonia, use of medication with a known dopaminergic or serotonergic effect during the study or in the 20 weeks preceding the baseline scans [18], and pregnancy or lactation. Neurological examination was normal except for dystonia, myoclonus and tremor in all patients. BoNT injections were administered at the first day of the medication trial or a maximum of 7 days prior/after starting the medication. Thus, baseline scans were acquired within a week of BoNT injections and follow-up scans were made between 5–7 weeks after BoNT injections.

### 2.2. Ethical Compliance

This study was approved by the Ethical Committee of the AMC (MEC10/036; approved date 31 May 2010), and informed consent was obtained from all patients. We confirm that we have read the Journal’s position on issues involved in ethical publication and affirm that this work is consistent with those guidelines.

### 2.3. Experimental Design and Treatment

Patients in the present imaging substudy participated in a larger randomized, double-blind, placebo-controlled, crossover trial, containing a total of 53 patients. For the details regarding randomization and treatment assignment, we refer to our previously published article [17]. In short, patients were randomly assigned to first receive one of two treatment options for the duration of six weeks: escitalopram 10 mg once daily, or placebo. Randomization per block of 4 subjects was performed and the treatment was blinded for both the patients and the investigator. The patients we report on in the current substudy are a subgroup that had baseline SPECT scans and neurological and psychiatric examination before start of the medication trial [7,8]. Medication was started immediately after all three baseline scans were performed (see Section 2.5). Both baseline scans ([^123^I]-FP-CIT (I-123 fluoropropyl carbomethoxy-3 beta-(4-iodophenyltropane)) and [^123^I]IBZM (I-123 iodobenzamide) SPECT; see Section 2.5) were acquired within 3 to 7 days of each other. The repeat SPECT scans (also made within 3 to 7 days of each other), as well as the neurological and psychiatric examinations were performed after a six-week treatment course with escitalopram or placebo. At the same time blood for analysis of plasma levels of escitalopram was withdrawn and stored at −20 °C until analysis. Samples were analysed in batches of 10–20 samples at a time using a validated LC-MS/MS method (liquid chromatography–mass spectrometry/mass spectrometry; range of detection 5–500 µg/L) [19]. The last dosage of medication had been less than 24 h before the scan for both scans. After a washout period of 2–6 weeks, interventions were switched. The halftime of escitalopram is 30 h, so a minimum period of 2 weeks was sufficient to washout escitalopram completely. Because of radiation burden, scans were not repeated after the second treatment round.

### 2.4. Scoring Neurological and Psychiatric Symptoms

Patients were systematically neurologically and psychiatrically examined, as published previously [7]. This examination was performed on the day the first SPECT scan was made and consisted of video recordings, a psychiatric interview and questionnaires. The detailed results of the effect of escitalopram and placebo on these examinations have been reported previously [17]. Considering the small sample size of the current study, we only incorporated the Clinical Global Impression Scale (CGI) [20], which is a seven-point scale of disease severity were lower scores indicate less symptoms. This question was answered for jerks, psychiatric symptoms and dystonia separately, both objectively by the physician and subjectively by the patient. We used the CGI to classify patients as responders (defined as a decrease of ≥1 point on CGI compared to baseline) or nonresponders (CGI same or higher compared to baseline). This was done separately for the three symptoms, and separately by patient and physician.

### 2.5. SPECT Imaging

All patients received 300 mg potassium iodide to block thyroid uptake of free radioactive iodide before administration of the tracer. Both striatal DAT and diencephalon/midbrain SERT were imaged using [^123^I]FP-CIT SPECT, a technique that has been validated before [21,22]. Subjects received a mean dose of 100 MBq of [^123^I]FP-CIT intravenously (produced according to GMP criteria by GE Healthcare) as a bolus [23]. Scans were performed 2 h after bolus injection to visualize and quantify the SERT binding in diencephalon/midbrain and 3 h after bolus injection to visualize and quantify the DAT binding in the striatum [24]. To visualize striatal D2/3R binding, subjects received a 56 MBq bolus of [^123^I]IBZM intravenously (produced according to GMP criteria by GE Healthcare) followed by continuous infusion of 14 MBq/h of [^123^I]IBZM until the end of the scan to achieve unchanging regional brain activity levels [25,26]. Acquisition of the images was started 2 h after the bolus injection [25,27]. All SPECT studies were performed on a 12-detector single-slice brain-dedicated scanner (Neurofocus 810, which is an upgrade of the Strichmann Medical Equipment) with a full-width at half-maximum resolution of approximately 6.5 mm, throughout the 20-cm field-of-view. After positioning of the subjects with the head parallel to the orbitomeatal line, axial slices parallel and upward from the orbitomeatal line to the vertex were acquired in 5 mm steps. An average of 15 slices was acquired in a 64 × 64 matrix. Scanning time was 3.5 min per slice for [^123^I]FP-CIT and 5 min per slice for [^123^I]IBZM SPECT studies. The energy window was set at 140–178 keV. Images were reconstructed in 3-D mode and analysed blindly by one observer (EZ). To quantify extrastiatal SERT, fixed regions of interest (ROIs) for the diencephalon and midbrain combined were positioned as earlier described [22]. To quantify striatal DAT, fixed ROIs for striatum were positioned as previously described [28]. The cerebellum was used as reference region for diencephalon/midbrain SERT binding as well as for striatal DAT binding [22]. For the [^123^I]IBZM images, fixed ROIs for the striatum were positioned on the occipital cortex on the same slices that were used as reference regions [29]. In all cases, specific to nonspecific binding ratios were calculated as [(activity in ROI–activity in reference region)/activity in reference region], representing the binding potential (BP_ND_) [30]. BP_ND_ is a combined measure of the density of available neuroreceptors/transporters and tracer affinity to the neuroreceptor/transporter. SERT occupancy was calculated as (BP_ND_ at baseline-BP_ND_ after escitalopram)/BP_ND_ at baseline * 100.

### 2.6. Statistical Analysis

The Mann–Whitney U test was used to calculate differences in BP_ND_ between different groups at baseline and Wilcoxon matched pairs signed rank-sum t-test was used to calculate differences in BP_ND_ before and after treatment. Spearman’s correlation was used to examine a possible correlation between escitalopram plasma levels and SERT occupancy. Analyses were carried out using SPSS (statistical package for the social sciences; version 24 and differences were considered significant at *p* < 0.05.

## 3. Results

In total, before- and after-treatment SPECT scans were acquired in 10 patients treated with escitalopram and in 8 patients treated with placebo. Despite indicating that they took all medication, two patients in the escitalopram group had nonmeasurable plasma levels and were excluded from the remainder of the analyses. Analyses were performed using the remaining 8 patients. As expected, none of the patients in the placebo group had measurable plasma levels of escitalopram. Baseline characteristics are depicted in Table 1. At baseline, 11/16 patients (69%; 5 in the placebo group and 6 in the escitalopram group) fulfilled the criteria for a psychiatric disorder. In total 8/16 patients fulfilled the criteria for an anxiety disorder (50%; 3 in the placebo group and 5 in the escitalopram group) and 3/16 patients fulfilled the criteria for a mood disorder (19%; 1 in the placebo group and 2 in the escitalopram group). There were no significant differences in baseline extrastriatal SERT, striatal DAT or striatal D2/3R BP_ND_ between patients randomized to be treated with escitalopram first and patients randomized to be treated with placebo first.

### 3.1. Difference between Baseline Scans and Scans after Treatment

There was no significant difference in extrastriatal SERT, striatal DAT or striatal D2/3R BP_ND_ between baseline and treatment with placebo. In patients treated with escitalopram, SERT BP_ND_ and DAT BP_ND_ were lower and D2/3R BP_ND_ was higher after treatment compared to baseline. However, the variability of the measurements was high, and results failed to reach statistical significance (Table 2).

### 3.2. Differences between SSRI and Placebo

There was a trend towards a lower median diencephalon/midbrain SERT BP_ND_ in patients treated with escitalopram (0.05 (IQR(interquartile range) −0.02−0.28) compared to patients treated with placebo (0.20 (IQR 0.08−0.39); *p* = 0.13 (see Figure 1)). Median SERT occupancy was 64.6% in the escitalopram group (IQR 1.2−91.9%). There was no significant difference in DAT BP_ND_ (*p* = 0.40) or D2/3R BP_ND_ (*p* = 0.46) between the treatment groups.

After treatment with escitalopram, patients that indicated on the CGI that dystonia improved (n = 6) had a trend towards lower extrastriatal SERT BP_ND_ on the post-treatment scan compared to patients (n = 2) that indicated that severity of dystonia was the same or worse (0.03 (IQR −0.04−0.10) vs. 0.33 (IQR 0.29-NA); *p* = 0.07). The median occupancy of SERT was 71.6% in patients that indicated dystonia improved on the CGI. SERT was not occupied in patients that did not indicate improvement (*p* = 0.14).

Patients (n = 5) that indicated on the CGI that their psychiatric symptoms (depressive symptoms and anxiety) improved after treatment with escitalopram had significantly lower SERT BP_ND_ on the post-treatment scan compared to patients (n = 3) that indicated that severity of psychiatric symptoms was the same or worse (0.01 (IQR −0.04−0.05) vs. 0.29 (IQR 0.25-NA); *p* = 0.04). The median occupancy of SERT by escitalopram was 76.5% in patients that indicated psychiatric symptoms improved, and SERT was less occupied in patients that did not indicate improvement (median occupancy −9%; *p* = 0.04).

In the escitalopram group, there was no significant difference in extrastriatal SERT BP_ND_ or occupancy between patients who indicated they did or did not improve in terms of jerks. There was no difference in DAT BP_ND_ and D2/3R BP_ND_ between escitalopram-treated patients who indicated they improved on dystonia, jerks or psychiatric symptoms compared to patients who did not improve. There was also no difference in SERT, DAT and D2/3R BP_ND_ between escitalopram-treated patients who did and did not improve in terms of dystonia, jerks or psychiatric symptoms according to the physicians.

Median plasma level of escitalopram was 21.0 µg/L (IQR 10.9−25.8). There was no correlation between SERT occupancy and plasma levels of escitalopram (correlation coefficient 0.21, *p* = 0.61). There was also no correlation between SERT BP_ND_ either pre- or post-treatment and plasma levels of escitalopram.

In the placebo group, there were no differences in SERT, DAT and D2/3R BP_ND_ between patients who did and did not improve in term of dystonia, jerks or psychiatric symptoms according to either the patients or the physicians.

## 4. Discussion

In our study, we found no significant difference in SERT, DAT and D2/3R BP_ND_ before and after treatment with either escitalopram or placebo. However, we did find a trend towards a difference in SERT BP_ND_ after treatment with escitalopram compared to placebo. Furthermore, after treatment with escitalopram, we found a significant higher SERT occupancy in patients who reported an improvement of psychiatric symptoms compared to those who did not. A trend towards a significant higher SERT occupancy was found in patients who reported an improvement of dystonia compared to those who did not.

The trend towards a lower SERT BP_ND_ in the escitalopram group represents occupancy of SERT of 65%. The lack of a statistically significant occupancy of SERT after a six-week treatment course was likely due to a combination of factors. One factor was low baseline extrastriatal SERT BP_ND_. Low baseline SERT binding is in line with previous human [^123^I]FP-CIT SPECT studies [22,31,32], and is related to the modest affinity of the radiotracer for the SERT [33]. The BP_ND_ levels did reduce to around 0 after treatment (see Table 2), reflecting substantial SERT occupancy, but the absolute difference in BP_ND_ was small. Another factor was that in our study the last tablet of escitalopram was taken the day before the scans were made. In healthy subjects using 10 mg escitalopram once daily for 10 days, SERT occupancy in the midbrain 6 and 54 h after ingestion of the last tablet showed a decrease from 82% after 6 h to 63% after 54 h [14]. It is likely that this effect also contributed to the relatively low SERT occupancy we found. However, our present study was not designed to test whether [^123^I]FP-CIT SPECT can be used to measure SERT occupancy by an SSRI, as this has been proven in several other experimental studies [12,31], but to evaluate the relationship between SERT occupancy and clinical effects. Nevertheless, in future studies focussing on SERT occupancy by an SSRI in dystonia, it might be better to assess the occupancy shortly after (supervised) administration of the last tablet.

There was no significant relationship between SERT occupancy and plasma levels of escitalopram, likely due to the ceiling effect where maximum SERT occupancy of around 80% is reached at relatively low dosages of SSRIs [13,31].

We believe that the difference in SERT BP_ND_ after citalopram presents predominantly occupancy of the SERT by the SSRI escitalopram. Indeed, in a previous [^123^I]FP-CIT SPECT study in healthy volunteers, we demonstrated that acute oral administration of the SSRI paroxetine can significantly block extrastriatal [^123^I]FP-CIT binding.^31^ In the present study, however, the subjects were treated for six weeks. Therefore, we cannot exclude that on top of the occupancy of the SERT by citalopram, also compensatory molecular mechanism has occurred, e.g., downregulation of the SERT, which might have slightly influenced the determination the SERT occupancy.

The most surprising, but interesting, finding of our study was that after escitalopram treatment patients, with high SERT occupancy reported a better clinical effect on dystonia and psychiatric symptoms compared to patients with low SERT occupancy. This effect was defined as a decrease of ≥1 point on self-reported CGI severity scores and there was no difference in an objective physician CGI score of disease severity. Because of the small sample size, we did not incorporate scores from depression or anxiety questionnaires, such as the Beck Depression Inventory or Beck Anxiety Inventory. However, our finding indicates that in patients whereby most SERTs are blocked by escitalopram experience a larger subjective treatment effect. Such a relationship between SERT occupancy and treatment effect has not been demonstrated in patients with major depression [16]. However, in patients with obsessive-compulsive disorder using sertraline, another SSRI, a positive relationship was detected between pre-treatment/baseline SERT BP_ND_ measured with [^123^I]β-CIT SPECT in the thalamus and hypothalamus and higher SERT occupancy, as well as better treatment response [34]. This is of interest since sertraline is the SSRI with the highest affinity for the DAT, thus an effect of SERT occupancy on treatment response can be debated. An alternative explanation for our finding of higher levels of SERT occupancy in patients with a positive subjective response is that increased SERT occupancy led patients to provide more positive answers. This was previously shown in patients with major depression and severe pessimistic attitudes who had low extracellular serotonin levels. These levels could be raised by the serotonin releaser d-fenfluramine, which led to more optimism in these patients [35]. A pessimistic attitude at baseline could have led to a high score on the CGI severity scale given by the patients and a shift towards optimism would have made the patients give themselves a better score. This might explain why there was a relationship between SERT occupancy and the positive subjective scores of the patients, but a lack of such a relationship between SERT occupancy and the objective physician scores.

In contrast with our hypothesis, no significant differences were found in striatal DAT BP_ND_ after treatment with escitalopram. Although not significant, the median DAT BP_ND_ we found in this current study was 14% lower after treatment with escitalopram compared to baseline. From studies in patients with major depression, we know that SSRIs have an effect on the dopaminergic system, but the decrease in DAT BP_ND_ is much smaller (around 20%) than the decrease in SERT BP_ND_ (40−70%). Furthermore, the decrease in DAT BP_ND_ is smaller in older patients [11]. There are also indications that the decrease in DAT BP_ND_ is related to the used reference region, at least when using nonselective tracers like [^123^I]FP-CIT. When using the occipital cortex as reference tissue, a larger effect on DAT BP_ND_ was found than when using the cerebellum, probably because of a higher concentration of SERTs in the occipital cortex. Other factors that can explain why some studies found an effect of SSRIs on striatal DAT BP_ND_ while others did not are intravenously given medication versus oral ingestion, chronic treatment with SSRIs versus acute administration of a single dose, and perhaps the cohort under study (patients like CD versus healthy controls may show a different neurotransmitter response).

We also did not find significant changes in D2/3R BP_ND_ after treatment with escitalopram. In patients with major depression, overall no treatment effect on D2/3R BP_ND_ was found using [^123^I]IBZM SPECT, but there were indications of a relationship between D2/3R BP_ND_ before and after treatment with either the SSRI paroxetine or fluoxetine. Patients that responded well to SSRIs had lower D2/3R BP_ND_ at baseline and D2/3R BP_ND_ increased after treatment [36]. We did not find a correlation between D2/3R BP_ND_ and treatment response, neither for the objective scores of the physicians, nor for the patients’ subjective scores.

This study has some limitations. The most important limitation is the sample size. This is especially important when measuring extrastriatal SERT BP_ND_ with [^123^I]FP-CIT SPECT of which it has been established that the BP_ND_ is lower with a larger range compared to striatal DAT BP_ND_ (measured with [^123^I]FP-CIT SPECT) or D2/3R BP_ND_ (measured with [^123^I]IBZM SPECT) [22,37]. A higher variability makes it statistically more difficult to detect small differences. Furthermore, we used [^123^I]FP-CIT to image both DAT and SERT BP_ND_ in the same patients. [^123^I]FP-CIT is a nonselective DAT/SERT SPECT tracer that can reliably be used to measure SERT BP_ND_ in the diencephalon/midbrain but not in other relevant brain regions like the striatum. In addition, we had to exclude two patients from the escitalopram group because of nondetectable escitalopram plasma levels. This might indicate they did not take medication; however, we cannot exclude that they were so-called fast metabolizers. In a study where 8 healthy controls were treated with the SSRI paroxetine for 8 days, and all tablets were ingested under supervision, one patient had nondetectable plasma levels of paroxetine and was considered a fast metabolizer [38]. We could measure plasma levels of ≥5 µg/L and a plasma level of 5 µg/L has been proven to give an occupancy of >80% in the midbrain [13]. However, SERT occupancy in these particularly two patients was in the negative range (BP_ND_ was even somewhat higher after treatment compared to baseline; data not shown) making it unlikely that these patients had plasma levels just below the detection range and making it more likely that they did not take study medication. Lastly, here we only measure short-term effects with a treatment duration of six weeks. It is very possible that with a longer treatment duration, neuroadaptive changes will occur and results will be different. It would be interesting to investigate this in a larger study with a longer treatment duration, for example a prospective cohort of patients with CD that are started on an SSRI to reduce psychiatric symptoms and undergo SPECT scans before treatment and after 3–6 months of treatment.

In conclusion, we performed the first serial SPECT study in patients with CD treated with escitalopram. We found a trend towards lower SERT BP_ND_ and consequently higher SERT occupancy after treatment with escitalopram. Most interestingly, we found that within the group treated with escitalopram, patients who reported subjective improvement in dystonia and/or psychiatric symptoms had significantly lower SERT BP_ND_ and higher SERT occupancy than patients who did not improve.

## Figures and Tables

**Figure 1 biomolecules-10-00880-f001:**
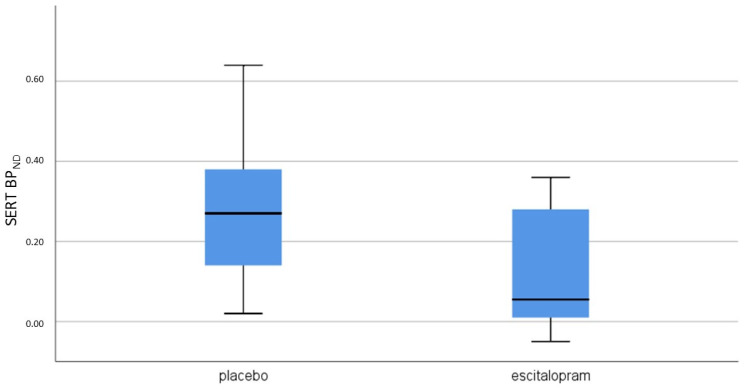
Median SERT BP_ND_ and interquartile range after treatment with placebo and escitalopram.

**Table 1 biomolecules-10-00880-t001:** Baseline Characteristics.

Variables	Escitalopram (n = 8)	Placebo (n = 8)	*p*-Value
Age, mean (SD)	56.4 (9.1)	56.8 (10.3)	0.94
Male, n (%)	4 (50%)	6 (75%)	0.61

n = number, SD = Standard deviation.

**Table 2 biomolecules-10-00880-t002:** Comparison of SPECT Measures (BP_ND_) between Pre- and Post-Treatment.

	Baseline	After Treatment	*p*-Value
Escitalopram (n = 8)			
SERT	0.18 (0.12−0.31)	0.05 ((−0.02)−0.28)	0.26
DAT	3.31 (2.47−4.10)	2.90 (2.65−3.97)	0.40
D2/3R	0.71 (0.60−0.92)	0.89 (0.50−0.92)	0.60
Placebo (n = 8)			
SERT	0.28 (0.16−0.38)	0.20 (0.08−0.39)	0.40
DAT	3.54 (3.40−3.62)	3.71 (3.34−4.56)	0.61
D2/3R	0.92 (0.62−1.13)	0.88 (0.84−1.03)	0.73

Extrastriatal SERT and striatal DAT were measured with [^123^I]FP-CIT SPECT, and striatal D2/3R with [^123^I]IBZM SPECT. Data are depicted as median (interquartile range).

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
