# Peer review of "The Effect of Escitalopram on Central Serotonergic and Dopaminergic Systems in Patients with Cervical Dystonia, and Its Relationship with Clinical Treatment Effects: A Double-Blind Placebo-Controlled Trial"

_biomolecules, 2020, doi:10.3390/biom10060880_

Round 1

Reviewer 1 Report

The authors improve paper and solve concerns.

Reviewer 2 Report

In the revised version of the manuscript, the Authors have addressed the requests, providing further details and elucidating unclear aspects of the study, along with appropriate changes. The quality of the manuscript has certainly improved.

This manuscript is a resubmission of an earlier submission. The following is a list of the peer review reports and author responses from that submission.

Round 1

Reviewer 1 Report

This manuscript focus on changes in dopaminergic and serotoninergic systems, as assessed by serial SPECT scans, in patients with cervical dystonia after a 6-week treatment with escitalopram or placebo. The authors hypothesized that, after escitalopram treatment, SERT binding would be reduced (representing occupancy), whereas D2/D3 receptor and DAT binding would be increased. In addition, they investigated whether changes in SPECT measurements were related to clinical outcomes. No differences in striatal D2/D3 receptor, striatal DAT and extrastriatal SERT binding were found in patients treated with escitalopram (n=8) or placebo (n=8). However, when comparing SPECT measurements after escitalopram to those after placebo, the authors observed a trend towards a lower extrastriatal SERT binding in the escitalopram group (median SERT occupancy of 65%). Interestingly, within the escitalopram group, patients reporting a subjective improvement in dystonia (n=6) or psychiatric symptoms (n=5) showed lower SERT binding and higher SERT occupancy in comparison to those who did not report an improvement on the CGI (n=2 for dystonia; n=3 for psychiatric symptoms). There were no differences in SPECT measurements between escitalopram treated patients who did and did not improve according to the physicians. 

The topic of this manuscript is interesting, particularly because dopamine and serotonin dysfunctions are hypothesized to play a relevant role in the pathophysiology of CD. Most importantly, neuroimaging studies investigating both the dopaminergic and serotoninergic system in CD, as well as the effect of additional or alternative treatment, are scarce.

Despite the appropriate design and originality of the study, the main limitation, as rightly stated by the authors, is the sample size. In my opinion, this makes it more difficult to interpret some of the results, particularly those obtained within the small escitalopram group, as well as to draw definite conclusions. Several issues/points should be addressed and/or clarified. 

My comments are: 

Material and Methods

-Although the authors refer to their previous studies in "Subiects", "Scoring neurological and psychiatric symptoms" and "Experimental design and treatment" sections (References 7, 8 and 17), for a better understanding, it is advisable to rearrange the above-mentioned sections, as they currently do not appear clear/linear enough.

-The term “idiopathic” should be added to CD (lines 80-81), according to the previously published studies. 

-Concerning neurological and psychiatric evaluations, CD patients underwent video-recordings, a psychiatric interview and questionnaires, as reported by the authors (line 98). Did the patients receive a depressive or anxiety disorder diagnosis at baseline? What was the amount of time elapsing from neurological and psychiatric evaluations to SPECT imaging? Please, address these points.

-The authors stated that considering the relatively small sample size of the current study, they only incorporated the Clinical Global Impression Scale…(lines 100-101). It would be advisable to remove the term “relatively”, as statistical analysis included 8 patients treated with escitalopram and 8 patients with placebo. 

-As regards SPECT imaging (line 127), the time elapsing from 123I-FP-CIT to 123I-IBZM scans is not mentioned the text. 

-Statistical tests used to assess differences in baseline characteristics (Table 1) should be added to the text (lines 158-162). 

-It would be advisable to add "escitalopram" to plasma levels (line 161). 

Results

-Additional clinical information (psychiatric comorbidity) should be reported in Table 1 (line 172).

-Concerning the differences between SSRI and placebo groups (lines 185-189), the authors stated that there was a trend towards a lower median diencephalon/midbrain SERT BPND in patients treated with escitalopram compared to patients treated with placebo. However, the p value (p=0.13) does not suggest a clear trend towards a difference. It would be worthwhile to include a Figure (comparison graphs) for a ready understanding. 

-Did the authors test for baseline differences in SPECT measurements between escitalopram and placebo groups? Please, address this point.  

-It would be advisable to replace “strong trend” with “trend” (line 191), considering the small number of patients included in the statistical analysis (6 vs. 2 patients). 

-As stated by the authors, SERT was less occupied in patients that did not indicate improvement (linee 202-203). By what extent?

-Regarding jerks (lines 205-206), in the escitalopram group there was no significant difference in extrastriatal SERT binding or occupancy between patients who indicated they did or did not improve. However, as reported in Table 1, only 4 patients suffered from jerks. How did the authors make the comparison? 

-It would be advisable to add more details (e.g. p value) resulting from the comparison between escitalopram treated patients who did and did not improve on dystonia and psychiatric symptoms according to the physicians (lines 209-210). 

Discussion

-It is advisable to remove the term “strong” in the sentence “We did find a strong trend towards a difference in SERT BPND after treatment with escitalopram compared to placebo” (lines 223-224). 

-Please, clarify the sentence “This effect was defined as an increase on self-reported CGI severity scores…” (lines 250-251). According to the Clinical Global Impression scale, lower scores indicate less severe symptoms. As reported in Material and Methods, a decrease of ≥ 1 point on GCI compared to baseline was used to classify patients as responders, whereas non-responders had the same or higher CGI compared to baseline (the CGI was scored by patients and physicians separately). Did the authors refer to an improvement (a decrease) on self-reported CGI severity scores? 

-The authors stated that BDI or BAI scores were not incorporated in the analysis because of the small sample size (lines 252-253). However, it would have been interesting to perform an additional analysis using more specific measurements (lines 252-253).  

-The authors hyphotesized that a pessimistic attitude at baseline could have led to a low score given by the patient and a shift towards optimism would have made the patients give themselves a better score (lines 266-267). Did they mean a worse (or high) score at baseline? Please, clarify. 

-The lack of significant differences in SPECT measurements before and after treatment should be mentioned (linee 317-322).

Reviewer 2 Report

It seems that it is a well-designed study to test if the mechanism of Citalopram on CD is related to its ability to bind SERT and additionally produce changes in DAT and D2/D3R occupancy and its correlation with clinical improvement. The main concern in this paper is the lack of statistical significance between, at least for the primary variable studied. That has a tendency of significance to the SERT BPND that are discussed in terms of possible explanations.

Discussion is limited and does not justify the publication based on a result of trend and a fascinating clinical finding that can not be appropriately linked. Some questions arose from a discussion that requires significant improvement.

If CIT has a modest affinity to bind SERT, why use it?

Although the study is not designed to evaluate SERT occupancy by drug, concerns of their use should be taken in mind during study design. The paragraph should be removed. In any case, the design of drug administration was the responsibility of the researcher.

Drug occupancy can be explained in terms of affinity and the number of receptors. Which one changes?

What does explains the tendency of changes?

The reported clinical effect of the drug is so significant and merits a better study design, with more patients that include inventories of depression and anxiety. Sure SPECT measures will give statistical significance.

It seems that it is a well-designed study to test if the mechanism of Citalopram on CD is related to its ability to bind SERT and additionally produce changes in DAT and D2/D3R occupancy. The main concern in this paper is the lack of statistical significance between, at least for the primary variable studied. That has a tendency of significance to the SERT BPND that are discussed in terms of possible explanations.

Discussion is limited and does not justify the publication based on a result of trend and a fascinating clinical finding that can not be appropriately linked. Some questions arose from a discussion that requires significant improvement.

If CIT has a modest affinity to bind SERT, why use it?

Although the study is not designed to evaluate SERT occupancy by drug, concerns of their use should be taken in mind during study design. The paragraph should be removed. In any case, the design of drug administration was the responsibility of the researcher.

Drug occupancy can be explained in terms of affinity and the number of receptors. Which one changes?

What does explains the tendency of changes?

The reported clinical effect of the drug is so significant and merits a better study design, with more patients that include inventories of depression and anxiety. Sure SPECT measures will give statistical significance.